# β-Catenin and SOX2 Interaction Regulate Visual Experience-Dependent Cell Homeostasis in the Developing *Xenopus* Thalamus

**DOI:** 10.3390/ijms241713593

**Published:** 2023-09-02

**Authors:** Juanmei Gao, Yufang Lu, Yuhao Luo, Xinyi Duan, Peiyao Chen, Xinyu Zhang, Xiaohua Wu, Mengsheng Qiu, Wanhua Shen

**Affiliations:** 1Zhejiang Key Laboratory of Organ Development and Regeneration, College of Life and Environmental Sciences, Hangzhou Normal University, Hangzhou 311121, Chinamsqiu@hznu.edu.cn (M.Q.); 2College of Life and Sciences, Zhejiang University, Hangzhou 310058, China

**Keywords:** thalamus, neurogenesis, tissue homeostasis, visual deprivation, β-catenin, *Xenopus*

## Abstract

In the vertebrate brain, sensory experience plays a crucial role in shaping thalamocortical connections for visual processing. However, it is still not clear how visual experience influences tissue homeostasis and neurogenesis in the developing thalamus. Here, we reported that the majority of SOX2-positive cells in the thalamus are differentiated neurons that receive visual inputs as early as stage 47 *Xenopus*. Visual deprivation (VD) for 2 days shifts the neurogenic balance toward proliferation at the expense of differentiation, which is accompanied by a reduction in nuclear-accumulated β-catenin in SOX2-positive neurons. The knockdown of β-catenin decreases the expression of SOX2 and increases the number of progenitor cells. Coimmunoprecipitation studies reveal the evolutionary conservation of strong interactions between β-catenin and SOX2. These findings indicate that β-catenin interacts with SOX2 to maintain homeostatic neurogenesis during thalamus development.

## 1. Introduction

In the central nervous system, functional neural circuits are developmentally constructed through the continuous proliferation and differentiation of neural stem/progenitor cells (NSCs/NPCs). A variety of factors, including multiple extracellular matrix components, cytoplasmic factors, and nuclear receptors, have been identified to regionalize the proliferation and differentiation of nerve cells in a spatial and temporal manner [1]. Comparative studies have revealed the conservation of molecular, structural, and functional characteristics of the thalamus in chicken, mouse, and *Xenopus* [2,3,4,5]. Although the visual experience has been shown to refine circuit connectivity, cell homeostasis, and synapse formation [6,7,8,9,10,11,12], the underlying mechanism by which visual experience induces the formation of the developing thalamus is less well understood.

In the visual system, sex-determining region Y (SRY)-related HMG box 2 (SOX2) is usually considered to be a persistent marker of multipotent neural stem cells, which plays important functions in progenitor cells’ development [13,14,15,16,17]. The SOXB1 subfamily comprised SOX1, SOX2 and SOX3 in mammals. SOX9 and SOX10 are classified within the SOXE group, which defines proliferating crest progenitors during embryogenesis [18]. SOX2 is a conserved transcription factor, which is required for neural differentiation [19] and progenitor cell maintenance [18,20,21,22]. The widespread expression of SOX2 plays region-specific and cell-type specific functions in different regions of developing and postnatal brain [17]. SOX2 deficiency results in impaired neurogenesis and pathological phenotype in humans [5]. We previously have shown that SOX2-positive cells distributed along the ventricles are brain lipid-binding protein (BLBP)-positive progenitor cells in the *Xenopus* tectum [14]. Although the roles of SOX2 in the thalamus are being investigated [23,24,25], little is known about SOX2 signaling and its functional involvement in visual experience-dependent thalamic development.

The expression of 22 markers, including Wnt3a, in the prepatterning and patterning of the *Xenopus* thalamus reveals a basic organization of the diencephalic region across vertebrates [26]. Wnt signaling is an evolutionarily conserved signal pathway that controls the balance of cell fate via the transcriptional coactivator β-catenin during brain development [27,28,29,30,31,32,33,34]. The phosphorylation of β-catenin is inhibited by the APC/Axin/CK1/glycogen synthase kinase 3β (GSK3β) destruction complex, resulting in β-catenin accumulation in the cytosol and translocation into the nucleus [35,36,37,38,39]. Activation of the canonical Wnt/β-catenin pathway leads to the transcriptional stimulation of lymphoid enhancer factor/T cell factor (LEF/TCF) target genes that modulate cell proliferation [40]. The target disruption of Wnt proteins or their receptors, such as LRP6, Axin2, and Frizzled3/5, results in severe deficits in thalamic development and disorders [41,42,43,44]. However, the functional role of β-catenin in stem cell self-renewal and tissue homeostasis has been largely debated, which is partly due to its capacity to form complexes with many downstream factors [45]. For instance, Wnt/β-catenin signaling may interact with the SOXB1, PAX6, OCT4 and SOX9 to maintain the regulatory networks for self-renewal and differentiation [13,46,47,48].

The thalamus serves as a relay station for sensory perception, movement, and cognitive functions through retina–thalamus–cortex connectivity [25,49]. In subcortical regions, such as the retina, superior colliculus, and thalamus, the influence of experience on circuit development is still controversial [50,51,52]. Monocular deprivation reduces the driving input, leading to alterations of thalamocortical projections and experience-dependent circuit refinement [52,53,54]. However, initial retinotopic map formation is largely dependent on spontaneous activity rather than visual experience [50]. Here, we took advantage of the *Xenopus* model system in live imaging and in vivo recording. We identified a subpopulation of SOX2-expressing neurons in the developing *Xenopus* thalamus. Visual deprivation shapes the morphology of thalamic neurons through retinal–tectal–thalamic connectivity. The functional interactions of β-catenin and SOX2 when visual deprivation is induced increase in proliferation and decrease in differentiation in the developing thalamus. Our findings indicated that the factors of β-catenin and SOX2 are evolutionarily conserved core components that synergistically control the thalamic homeostasis through physical interactions in vivo.

## 2. Results

### 2.1. SOX2 Expression Is Highly Regionalized in the Developing Thalamus

We previously showed that the SOX2 immunoreactive (SOX2^+^) cells located along the midline of the ventricle are BLBP^+^ radial glial cells that function as progenitor cells in the developing *Xenopus* tectum [14]. To fully characterize the precise topological distribution of SOX2 expression, we performed the whole-mount immunostaining of a tadpole using an anti-SOX2 antibody (Figure 1A). The coronal (Figure 1C) and sagittal cryostat sections (Figure 1D) showed that SOX2^+^ cells were widely distributed with distinct populations in the nervous system, including the olfactory bulbs (OB), rhombencephalon (Rho), optic tectum (OT), and thalamus (Th). Several markers including SOX2 and Nkx2.2 were specifically involved in thalamic development [26]. We observed that Nkx2.2 was expressed adjacent to SOX2 within the thalamus without any overlapping (Figure 1B). The combination of Nkx2.2 and SOX2 allowed for the identification of subpopulations of two discrete thalamic clusters (rostral and caudal thalamus) [15,26]. SOX2 expression in the thalamus first appeared at stage 34 and developmentally increased from stage 40 to stage 49 (Appendix A), indicating an enlargement of the thalamus in SOX2-expressing cells following brain development. The ratio of SOX2^+^ cells to thalamic cells (DAPI^+^) was ~25.3% at stage 40 and remained stable at ~32.1% from stage 46 to stage 49 (Appendix A), indicating a rapid increase in SOX2^+^ cells at the earlier stage. Notably, SOX2 (Figure 1B–D) but not SOX9 (Appendix A) expression was exclusively concentrated in the thalamus at stage 49. According to the distribution pattern of BLBP^+^ progenitor cells (Appendix A), expressions of SOX2 and SOX9 along the ventricle can be used as markers for proliferating cells. As a result, we reasoned that the representative SOX2 expression gradient in the thalamus might allow us to investigate the homeostatic regulation of thalamic cell number.

### 2.2. The Majority of SOX2^+^ Cells Are Differentiated Neurons in the Developing Thalamus

To characterize the cell identities of SOX2-expressing thalamic cells, we first evaluated BrdU incorporation by injecting BrdU (10 mM) and coimmunostaining brains with anti-BrdU and anti-SOX2 antibodies after 2 days. We observed that SOX2^+^ cells incorporated less BrdU (Figure 2A) without overlapping with PAX7^+^ cells in the thalamus (Appendix A). The discrete distribution patterns of SOX2/Nkx2.2, two thalamic markers, and PAX7, a regional marker [26], were used as a boundary to dissect the thalamus in this study. Furthermore, although BrdU^+^ and PH3^+^ cells are two distinct populations of progenitor cells in the thalamus (Appendix A), most BrdU^+^ cells are colocalized with PCNA^+^ cells (Figure 2B), indicating that BrdU can be used as a proliferating marker in the thalamus. Most thalamic SOX2^+^ cells are not colocalized with PCNA^+^ (Figure 2C), BLBP^+^ or vimentin^+^ cells (Appendix A) but are largely colabeled with neuronal markers of HuC/D and tubulin (Figure 2D,E), indicating that the great majority of thalamic SOX2^+^ cells are differentiated neurons. To substantiate the findings, we performed post-immunostaining for the thalamus with the anti-SOX2 antibody followed by pSOX2::GFP transfection, which was used to increase the cell-type specificity of progenitor cells in the tectum [55]. In agreement with the immunofluorescence data, the colocalized SOX2^+^ cells showed typical neuronal morphology with dendrites (Figure 2F), confirming that SOX2 is preferentially expressed in post-mitotic thalamic neurons.

### 2.3. Thalamic Neurons Receive Retinal Signals and Respond to Visual Deprivation

The optic tectum receives retinal signals and responds to visual deprivation (VD) by enhancing radial glia proliferation in the ventricle [7,56]. To functionally determine whether the developing thalamus responds to visual inputs, we placed tadpoles at stage 46 in a 12 h light/12 h dark or 24 h dark box for 48 h (Appendix A) and recorded the SOX2^+^ cells (Figure 2F) 100–200 µm distant from the ventricle layer using a patch–clamp setup (Appendix A). We measured the delay from the start of the light OFF stimulus to the onset of excitatory compound synaptic currents (eCSCs) in the neurons of the thalamus (Ctrl-Th) and the optic tectum (Ctrl-OT) (Figure 3A). We found that eCSCs delay was elongated in thalamic neurons compared to tectal neurons (Figure 3B). Visual deprivation dramatically increased the delay in VD-Th neurons compared to Ctrl-Th neurons (Figure 3B), implying that deprived visual inputs interfere with thalamic connectivity. We examined the integrated charge transfer of OFF stimuli–evoked eCSCs and found that the eCSCs were significantly smaller in Ctrl-Th thalamic neurons than those in Ctrl-OT tectal neurons (Figure 3C). Interestingly, the eCSCs in VD-Th neurons were significantly increased compared to those in Ctrl-Th neurons (Figure 3C), suggesting that VD may induce a homeostatic upregulation of synaptic currents. To test whether VD–induced changes in synaptic transmission result from the retina or the retino–thalamic visual pathway, we further recorded optic chiasm–induced excitatory postsynaptic currents (Figure 3D–F) and found that the changes are comparable to visual stimulation–induced eCSCs, suggesting that VD mainly alters the retino–thalamic inputs rather than the retina itself [57].

To test whether VD alters neuronal excitability, we measured the injected current–induced action potentials in the optic tectum and thalamus (Figure 3G). The number of action potentials in Ctrl-Th neurons was lower than in Ctrl-OT neurons. VD induced a considerable reduction in the number of spikes in VD-Th–treated neurons compared to Ctrl-Th neurons (Figure 3H), implying that VD may be able to alter the process of neural maturation. Furthermore, there were no significant differences in whole–cell capacitance, input resistances or resting membrane potentials (Appendix A).

To test whether VD changes neuronal morphology, we reconstructed the dendrites of pSOX2::GFP–transfected thalamic neurons (Figure 3I). We found that the total dendritic branch length (TDBL) and total branch tip number (TBTN) were significantly decreased in VD-treated thalamic neurons (VD-Th) compared to control thalamic neurons (Ctrl-Th) (Figure 3J,K). These results indicate that visual deprivation interferes with the retino–thalamic connections and thalamic neuronal morphology, allowing us to study visual experience–dependent thalamic neurogenesis in the developing brain.

### 2.4. Visual Deprivation Induces an Increase in Progenitor Cells and a Decrease in Differentiated Neurons

To determine whether VD affects the proliferation rate, we injected BrdU and immunostained the thalamus with anti-BrdU and anti-SOX2 antibodies (Figure 4A,B). The cryostat sections were collected, and all numbers of BrdU^+^ and SOX2^+^ cells (SOX9^−^ or BLBP^−^) in the thalamus were counted by confocal scanning (see methods). We found that BrdU^+^ cells were greatly increased but SOX2^+^ cells were significantly decreased in VD-treated tadpoles (Figure 4C–E). We used an anti-PH3 antibody to label mitotic cells and found that VD induced a significant increase in PH3^+^ cells (Appendix A). These data indicate that VD promotes cell proliferation while inhibiting cell differentiation in the developing thalamus. The changes in the BrdU^+^ and SOX2^+^ cell numbers in response to VD are comparable to those by collecting z–stack images for all cryostat sections (Figure 4F–H). To exclude the possibility that the decrease in SOX2 expression could be due to cell apoptosis, we performed a TUNEL experiment and found that VD decreased the number of apoptotic cells in the thalamus (Appendix A), indicating that the VD–induced decrease in SOX2^+^ neurons is not a result of cell apoptosis.

To further confirm whether VD alters the differentiation, the tadpoles were immunostained with anti-SOX2 and anti-HuC/D antibodies (Figure 4I,J). Both SOX2^+^ and HuC/D^+^ neurons were significantly decreased in VD–treated tadpoles compared to control tadpoles in the thalamus (Figure 4K,L). The colocalization and cell counting between SOX2 and HuC/D were confirmed by analyzing z–stack images with the Surphase module in iMaris (Appendix A). The VD–induced downregulation of HuC/D was reinforced by Western blotting, showing significantly lower expression of HuC/D in the VD-Th compared to the Ctrl-Th (Appendix A). These findings indicate that VD causes tissue homeostasis to shift toward proliferation rather than differentiation.

### 2.5. Visual Deprivation–Induced Homeostatic Regulation of Thalamic Cells Is Accompanied by Phosphorylation and Degradation of β-Catenin

The translocation of β-catenin into the nucleus is known to be involved in the regulation of cell proliferation and differentiation by the canonical Wnt/β-catenin signaling pathway. To determine the potential role of β-catenin in thalamic development, we immunostained the entire brain with anti-β-catenin and anti-SOX2 antibodies. At stage 49 *Xenopus*, we observed that β-catenin was restricted to the cytoplasm of most tectal cells but strongly expressed in the nuclei of SOX2^+^ thalamic cells (Figure 5A). VD exposure dramatically reduced the number of β-catenin nuclearized cells and SOX2^+^ cells (Figure 5B–D). This pattern of alterations corresponds to the lower levels of thalamic β-catenin and SOX2 expression in the VD–treated tadpoles compared to Ctrl tadpoles by Western blot (Figure 5E–G). The individual blots were shown in the Appendix A. The integrity of the dissected thalamus was confirmed by immunostaining with the anti-SOX2 antibody followed by thalamic ablation (Appendix A). In addition, the level of phosphorylated β-catenin (P-β-Cat at Ser33/37/Thr41) was considerably higher in VD-Th cells than in Ctrl-Th cells (Figure 5E,H), implying that β-catenin was phosphorylated and degraded following visual deprivation.

### 2.6. Wnt/β-Catenin Signaling Is Necessary and Sufficient to Mediate VD-Induced Thalamic Homeostasis

To test whether β-catenin and SOX2 affect thalamic development, we used antisense morpholinos of β-Cat-MO or SOX2-MO to downregulate either the β-catenin or SOX2 expression. The results revealed that SOX2-MO or β-Cat-MO effectively suppressed the endogenous expression of β-catenin or SOX2 (Figure 6A–C). The individual Western blots were shown in Appendix A. SOX2 expression was reduced when β-catenin was knocked down (Figure 6A–C), implying that β-catenin is required for SOX2 expression. Interestingly, SOX2 knockdown also reduced the expression of β-catenin (Figure 6A–C), suggesting that a potential feedback mechanism controls the SOX2 expression. To test whether β-catenin knockdown affects the proliferation and differentiation of thalamic cells, we transfected Ctrl-MO or β-Cat-MO into the brain (Figure 6D,E) and found that the knockdown of β-catenin increased the number of BrdU^+^ progenitor cells but decreased the number of HuC/D^+^ neurons (Figure 6F,G). The transfection of β-Cat-MO decreased the number of apoptotic cells in the thalamus (Appendix A), suggesting that the decrease in HuC/D^+^ cells is not attributable to cell apoptosis. These findings indicate that β-catenin is involved in the regulation of thalamic cell proliferation and differentiation.

We then investigated whether Wnt signaling activation could block VD–induced changes in thalamic cell proliferation and differentiation. First, we administered IWR-1-endo or TDZD-8 to tadpoles for 48 h and harvested brains to perform Western blotting analysis (Figure 7). We observed that IWR-1-endo significantly decreased β-catenin expression but TDZD-8 increased β-catenin expression (Figure 7B), indicating that β-catenin expression can be pharmacologically controlled in the brain. The VD–induced decrease in β-catenin was prevented by TDZD-8 treatment (Figure 7C,D). We also measured the changes in the phosphorylation of β-catenin. The VD-induced increase in β-catenin phosphorylation was reduced by TDZD–8 treatment (Figure 7E,F). These findings indicate that TDZD-8 stabilizes β-catenin by preventing it from being phosphorylated and degraded.

To further support the evidence of VD altering the neurogenesis by β-catenin signaling, we counted the total number of SOX2^+^ and β-catenin^+^ cells in TDZD-8–treated tadpoles (Figure 7G). In comparison to the Ctrl group, TDZD-8 alone resulted in an increase in SOX2^+^ and β-catenin^+^ cells (Figure 7H,I). The decrease in SOX2^+^ and β-catenin^+^ cells caused by VD was markedly blocked by TDZD-8 treatment (Figure 7H,I). The VD–induced increase in BrdU^+^ cells is prevented by TDZD-8 treatment (Appendix A). These results indicate that stabilizing β-catenin restores the VD–induced decrease in SOX2^+^ cells in the thalamus.

### 2.7. The Evolutionarily Conserved Crosstalk between SOX2 and β-Catenin

β-catenin and SOX2 have been shown to positively control each other’s expression (Figure 6A–C), raising the possibility that SOX2 may interact with β-catenin to modulate tissue homeostasis and neurogenesis. We dissected the brain and performed reciprocal coimmunoprecipitation assays for β-catenin and SOX2. β-catenin was precipitated from the resulting tissue lysates with an anti-SOX2 antibody, and the immunoprecipitates were then subjected to anti-β-catenin and anti-SOX2 Western blotting (Figure 8A). As a positive control, the whole lysates, marked as input, were blotted with anti-β-catenin and anti-SOX2 antibodies (Figure 8A). The homogenates immunoprecipitated with IgG alone were used as a negative control. The results revealed that endogenous β-catenin and SOX2 robustly interact with each other in the brain of control tadpoles (Figure 8A,B). These results reveal evidence of in vivo physical interaction between β-catenin and SOX2.

To test whether VD affects the interactions between β-catenin and SOX2, we immunoprecipitated with anti-SOX2 antibody and immunoblotted with anti-β-catenin antibody in control and VD–treated thalamus. We found that the intensity of β-catenin was significantly increased in VD tadpoles compared to control tadpoles (Figure 8C,D), suggesting that VD may stabilize the β-catenin and SOX2 complex to regulate the activation of gene transcription.

To gain insight into the evolutionary conservation of the β-catenin and SOX2 interaction, we subjected total protein extracts from the mouse thalamus (P12) to perform immunoprecipitations with anti-SOX2, and anti-β-catenin antibodies followed by Western blotting (Figure 8E,F). The results showed strong reciprocal interactions between β-catenin and SOX2, indicating molecular conservation of protein interactions between *Xenopus* and mouse during early thalamic development.

## 3. Discussion

In this study, we identified that the majority of SOX2^+^ cells in the developing thalamus are differentiated neurons. Importantly, we provide immunofluorescent, morphological and electrophysiological evidence showing that β-catenin nuclearized thalamic cells receive visual signals and exhibit a physiological response to visual deprivation as early as stage 49 *Xenopus*. The Wnt/β-catenin signaling pathway leads to visual deprivation–induced thalamic neurogenesis that favors the generation of progenitors over differentiated neurons, maintaining a balance between proliferation and differentiation homeostasis. We also show that the complex interactions between SOX2 and β-catenin regulate thalamic homeostasis in the *Xenopus* brain.

SOX2 is one of the SoxB1 subfamilies of HMG box transcription factors that maintain the proliferation of multipotent stem cells and act as a transcriptional repressor of neuronal target genes [18,20,58,59]. Here, we show two populations of SOX2–expressing cells in the *Xenopus* brain: largely differentiated neurons in the caudal thalamus [15] and BLBP–/SOX9–expressing progenitors in the ventricular layer. The members of the SOXB1 family are functionally redundant but not divergent *SOX* genes such as *SOX9* [20], implying that the differential distribution may contribute to this phenotypic consequence (Appendix A). These findings expand our knowledge of the multiple expression patterns of SOX2 in progenitor cells and differentiated neurons. The scattered BrdU^+^ and SOX2^+^ cells in the developing thalamus indicate that SOX2^+^ neurons may be derived in part from thalamic tissue progenitors, which is consistent with the studies showing the existence of a population of PH3–positive dividing progenitor cells in mouse thalamus [60] and our studies (Appendix A). The distinct expressions of SOX2, SOX9, PAX7 and Nkx2.2 support the idea that the thalamus generates distinct sets of thalamic nuclei in a spatial and temporal manner [61,62,63]. Further investigation using developmental lineage tracing may be required to determine the origination of thalamic cells [58].

Thalamic afferents into cortical plates influence cell proliferation and differentiation, in addition to the regionalization and specification of neocortical areas [52,64,65]. Given the importance of thalamic cortical connections, it is essential to understand whether thalamic cells are regulated by visual inputs. Thalamic neurons receive direct retinal inputs by morphologically identified retinal projections in developing tadpoles and adult *Xenopus laevis* [15]. Our in vivo electrophysiological recordings indicate that thalamic neurons receive afferent inputs from the retina, which responds to light ON and OFF as early as the tadpoles from stage 47, as shown previously in Zebrafish larva [66]. The evidence of longer delay and reduced dendritic length in thalamic neurons supports that visual experience may provide the spatial and temporal organization of local circuitry niche components to thalamic cell neurogenesis. In contrast, 48 h of visual deprivation was sufficient to reduce the expressions of β-catenin and SOX2, together with an increase in BrdU^+^ progenitors, indicating that visual experience–dependent tissue homeostasis depends on cell cycle exit regulation.

The canonical Wnt/β-catenin cascade has emerged as a critical regulator of thalamic development, connectivity, and diseases [17,40]. The expression of β-catenin protein starts as early as the egg stages, persists through gastrula stages, and accumulates in the nuclei on the dorsal side of the embryo [67,68,69,70,71]. We observed that β-catenin constitutively accumulates in the thalamic nucleus, which may be associated with the lack of a ubiquitination–dependent degradation pathway [72,73]. The deletion of β-catenin suppresses dorsal mesoderm induction and later the axon arborization of retinal ganglion cells in early *Xenopus* embryos [74,75]. The enhancement of β-catenin signaling increases the number of differentiated neurons in cultured ES cells [76,77] and mouse embryonic stem cell cultures [78]. We used a GSK-3β inhibitor of TDZD-8 or a Wnt signaling inhibitor of IWR-1-endo to upregulate or downregulate β-catenin activity in the *Xenopus* brain, as previously shown in the adult zebrafish [79]. The VD exposure–induced decrease in β-catenin nuclearized SOX2^+^ cells is significantly blocked by TDZD-8, indicating that the upregulation of β-catenin nuclearization is sufficient to increase the differentiation of SOX2^+^ neurons. The endogenous knockdown of β-catenin by a morpholino further confirms the shift of tissue homeostasis toward proliferation.

The synergistic action of a specific SOX2 partner is essential in mediating SOX2–dependent tissue homeostasis. For instance, SOX2 and Oct4 can bind directly to transcriptionally regulate ECS differentiation [80,81]. SOX3 and SOX17 have been shown to interact with β-catenin in vitro and repress Wnt gene expression [47,82], suggesting that SOX family members may antagonize Wnt signaling via β-catenin sequestration in osteoblast cells [83]. The knockdown of SOX2 prevents neural specification and differentiation in the *Xenopus* neural plate [19] but increases the number of differentiating neural cells in the chick spinal cord [21]. Our data support the idea that SOX2 synergizes with Wnt/β-catenin signaling to regulate thalamic cell proliferation and differentiation [22]. SOX2 knockdown decreases β-catenin protein levels, indicating that SOX2 could stabilize the β-catenin protein, which is required for activating Wnt/β-catenin and SOX2 target genes. Despite the key roles of SOX2 and β-catenin in the central nervous system, they have been studied separately in this context. Based on our immunoprecipitation results of the visual–experience–dependent strong interaction between β-catenin and SOX2, we reveal a potential self–reinforcing regulatory loop that maintains tissue homeostasis and circuit connectivity via the SOX2/β-catenin complex in the developing thalamus.

Accumulating evidence has shown that nuclearized β-catenin activates the transcription of TCF/LEF target genes [84,85], which controls cell fate determination and differentiation [86,87]. These observations imply that the canonical Wnt signaling pathway triggers TCF/LEF/β-catenin crosstalk to activate Wnt target genes, which may be repressed by the SOX2 protein [48]. The interaction between β-catenin and SOX2 proteins may be essential for maintaining the balance between the proliferation and differentiation of neural progenitor cells, as visual deprivation shifts toward more progenitor cells at the expense of differentiated neurons. Further analysis of Wnt target genes may help elucidate the molecular mechanism by which β-catenin expression levels maintain the homeostasis of thalamic development [27,28,36,72,87,88].

## 4. Materials and Methods

### 4.1. Animals

All animal procedures were performed according to the requirements of the ‘Regulation for the Use of Experimental Animals in Zhejiang Province’. Tadpoles were obtained by mating adult male/female *Xenopus laevis* injected with human chorionic gonadotropin (HCG) and raised on a 12 h dark/12 h light cycle in Steinberg’s solution [(in mM): 10 HEPES, 58 NaCl, 0.67 KCl, 0.34 Ca (NO_3_)_2_, 0.83 MgSO_4_, pH 7.4] within a 20 °C incubator. For experimental manipulations, tadpoles were anesthetized in 0.02% MS-222 (3-aminobenzoic acid ethyl ester methanesulfonate, Sigma Aldrich, St. Louis, MO, USA). The stages of tadpoles were characterized according to the developmental changes in the anatomy [89].

C57BL/6 laboratory mice were kept in a controlled environment with a regulated temperature of 22 ± 1 °C and a 12 h light/dark cycle. The mice’s overall health was monitored daily throughout the study [90]. For the experiments, the mice were humanely euthanized using Avertin–induced deep anesthesia. The thalamus tissue was carefully extracted in accordance with the mice brain map. All procedures involving animals adhered to ethical guidelines and were approved by both the Laboratory Animal Center and the Animal Ethics Committee of Hangzhou Normal University, China (permit number 2022–1063, issued on 3 March 2022).

### 4.2. Morpholinos and Transfection

To knock down endogenous protein expression, we used translation–blocking morpholinos (MOs, GeneTools) against *Xenopus* SOX2 (SOX2-MO, CGGTCTCCATCATGCTGTACAT) and *Xenopus* β-catenin (β-Cat-MO, TTTCAACCGTTTCCAAAGAACCAGG). The control tadpoles were transfected with a control MO (Ctrl-MO, GATGGCATGTCTCCTCGCCTTTGGA). All morpholinos were tagged with Lissamine for fluorescent visualization. The plasmid of pSOX2::GFP (0.5 μg/μL, gifted from Hollis Cline laboratory, Cold Spring Harbor, NY, USA) was used to visualize SOX2-positive cells in the thalamus. For whole–brain/thalamus electroporation, tadpoles were anesthetized, and morpholinos (10 μM) or plasmids (0.5 μg/μL) were injected into the midbrain ventricle/third cerebral ventricle. The two parallel platinum electrodes were placed on the skin above the tectum/thalamus, and current pulses were applied by a stimulator. The transfected tadpoles were screened for further experiments.

### 4.3. BrdU Labeling

BrdU (5-bromo-2-deoxyuridine, 10 mM, MP Biomedicals, Irvine, CA, USA) *w*/*o* morpholinos filled with a glass electrode was slowly injected into the third ventricle to observe the thalamic progenitor cells. The tadpoles were incubated in Steinberg’s solution for 2 days, which was followed by being anesthetized and fixed in PFA overnight at 4 °C. Brain sections were treated with 2 N HCl for 45 min at 37 °C to denature the DNA and rinsed 3 times. All sections were immunostained with anti-BrdU (1:100, Mouse, Sigma, B2531, St. Louis, MO, USA) antibody and secondary antibodies for image collection by a confocal microscope (LSM710, Zeiss, Oberkochen, Germany).

### 4.4. Immunohistochemistry and Image Analysis

Tadpoles were anesthetized and fixed in 4% paraformaldehyde (PFA, pH 7.4) at 4 °C overnight. Tadpoles were rinsed with 0.1 M phosphate buffer (PB, pH 7.4) and immersed in 30% sucrose overnight for dehydration. On the second day, animals were embedded in optimal cutting temperature (OCT) media and cut into 20 µm cryostat sections with a microtome (Microm, HM550 VP, Boise, ID, USA). Sections were rinsed with PB for 2 × 20 min, permeabilized with 0.3% Triton X-100 in PB for 4 × 10 min, and blocked with goat serum for 30 min before incubating with primary antibodies at 4 °C overnight. For primary antibodies, we used the antibodies of anti-SOX2, anti-SOX9, anti-HuC/D, anti-tubulin, anti-Nkx2.2, anti-β-catenin, anti-PCNA, anti-BLBP, and anti-vimentin (Table 1). Sections were rinsed with PB and incubated with secondary antibodies (FITC, Rhod, or Alexa 647) for 1 h at room temperature. After sections were counterstained with DAPI, mounted, and sealed, immunofluorescence images were collected using a confocal microscope. Immunopositive cells were counted from all 5 consecutive brain sections in each tadpole using the Surface module by iMaris 9.0 (Bitplane AG, Zurich, Switzerland) image processing software [91]. The number of immunoreactive cells from all sections was counted as the total cells from the entire brain or thalamus. All sections were prepared, imaged, and analyzed in parallel across samples. The boundary of the thalamus was determined by immunostaining with anti-SOX2, anti-PAX7 and anti-Nkx2.2 antibodies.

### 4.5. Whole–Mount Immunofluorescence

Fixed tadpoles were washed with 0.1 M PB for 2 × 20 min and permeabilized with 0.3% Triton (New York, NY, USA) X-100 in PB for 4 × 10 min. Tadpoles were placed in blocking buffer (3% normal goat serum in 0.3% Triton X-100) for 1 h and incubated for 2 days at 4 °C with an anti-SOX2 antibody (1:200, Mouse, Cell signaling, 4900, Danvers, MA, USA), which was followed by additional washes and detection with secondary antibodies for 2 days at 4° C. Embryos were washed and mounted for photography using a confocal microscope.

### 4.6. Immunoblotting

Animals were anesthetized in 0.02% MS-222. The brain was exposed by peeling off the covered skin. The dissected optical tecta or thalamus (approximately 30–50 brains for each group) were homogenized in radioimmune precipitation assay (RIPA) buffer with a protease inhibitor cocktail (Sigma Aldrich) and phenylmethylsulfonyl fluoride (PMSF, Solarbio, Beijing, China) at 4 °C. Protein concentrations were measured by BCA assay using a Nanodrop (Thermo Scientific, Waltham, MA, USA, 2000c). Protein homogenates were separated by SDS–PAGE (Bio-Rad, Hercules, CA, USA) and transferred to PVDF membranes. Membranes were blocked in 4% nonfat milk for 1 h with TBS buffer containing 0.1% Tween-20 (Sigma Aldrich) (TBST) and incubated with primary antibodies overnight at 4 °C. Antibodies were diluted in 1% nonfat milk. We used the following antibodies: anti-Phospho-β-catenin, anti-β-catenin, anti-SOX2, anti-HuC/D, and anti-GAPDH (Table 1). Blots were rinsed with TBST and incubated with horserace dish peroxidase (HRP)-conjugated secondary antibodies (goat anti-rabbit IgG (1:2000, CWbiotech, Beijing, China, CW0103), goat anti-mouse IgG (1:2000, CWbiotech, CW0102), goat anti-rabbit IgG heavy chain (1:2000, ABclonal, Woburn, MA, USA, AS063), goat anti-mouse IgG light chain (1:2000, ABclonal, AS062)) for 1 h at room temperature. Bands were visualized using ECL chemiluminescent (1:1, Pierce, Appleton, WI, USA).

### 4.7. Immunoprecipitation

The whole brains or thalamus were harvested and lysed in 100 µL of RIPA buffer with protease inhibitors for 1 h at 4 °C. Samples of 20 µL were taken from the lysate for the input control and mixed with 2 X sample buffer. The remaining 40 µL was added to protein A/G agarose beads (CWbiotech, CW0349S) for 3 h and incubated with 0.5–2 µg of specific antibody. The samples were collected by centrifugation, washed 3 times, boiled for 10 min, and subjected to Western blot analysis.

### 4.8. Electrophysiology

Tadpole preparation for patch clamp was performed as described previously [92]. All recordings were performed at room temperature (20–22 °C). For recordings in the thalamus, the superficial cells were removed by a suction pipette. The recording micropipettes were placed in the thalamus 100–200 μm away from the ventricle. Tadpoles were perfused with an external solution containing (in mM: 115 NaCl, 2 KCl, 3 CaCl_2_, 1.5 MgCl_2_, 5 HEPES, 10 glucose, and 0.01 glycine, pH 7.2, osmolality 255 mOsm). Excitatory postsynaptic currents were recorded by holding the membrane potential at −60 mV with intracellular solution containing (in mM: 110 K-gluconate, 8 KCl, 5 NaCl, 1.5 MgCl_2_, 20 HEPES, 0.5 EGTA, 2 ATP, and 0.3 GTP). Recording micropipettes were pulled from borosilicate glass capillaries and had resistances in the range of 9–11 MΩ. The liquid junction potential was adjusted during recording. Signals were filtered at 2 kHz with a MultiClamp 700B amplifier (Molecular Devices, Palo Alto, CA, USA). Data were sampled at 10 kHz and analyzed using ClampFit 10 (Molecular Devices, San Jose, CA, USA) or MiniAnalysis 6.03 (Synaptosoft, Fort Lee, NJ, USA).

### 4.9. Drugs and Treatment

The tadpoles were incubated with IWR-1-endo (10 μM, Selleck, Detroit, MI, USA, S7086, 10 mM stock in DMSO), a Wnt signaling inhibitor, or TDZD-8 (0.11 μM, Selleck, S2926, 10 mM stock in DMSO), a GSK-3β inhibitor [79]. If not stated otherwise, tadpoles were treated in Steinberg’s solution for 48 h.

### 4.10. Statistics

Two groups were tested with Student’s *t*–test. Unless noted, multiple group data were tested with ANOVA followed by a post hoc Tukey’s test. Data are represented as the mean ± SEM. Experiments and analysis were performed blind to the experimental conditions unless noted.

## Figures and Tables

**Figure 1 ijms-24-13593-f001:**
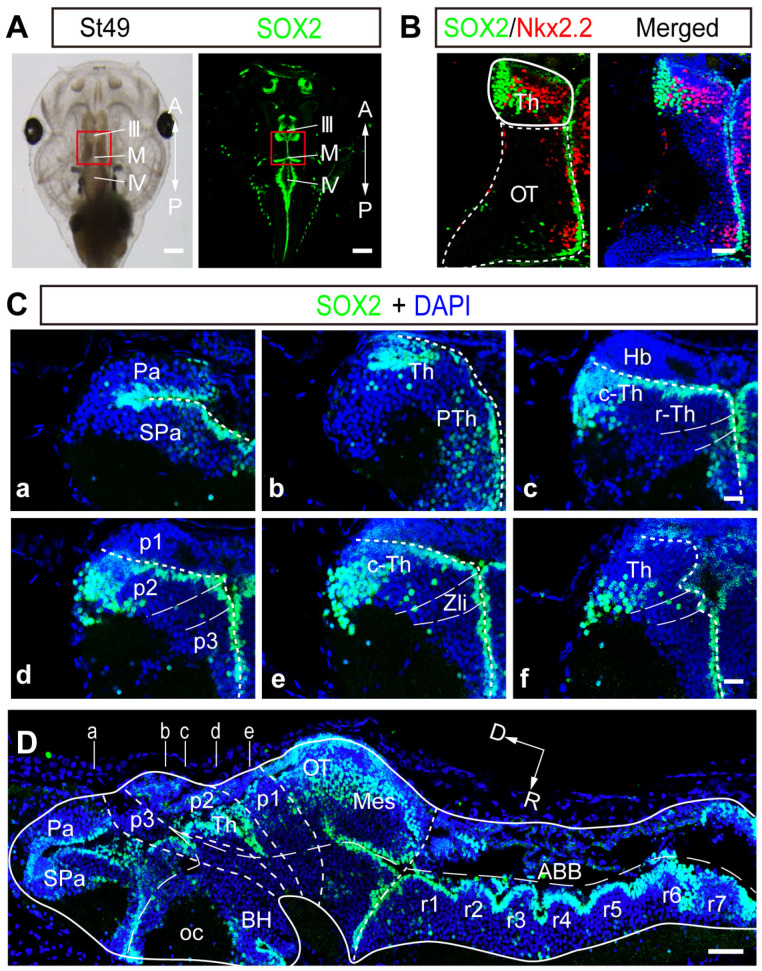
**SOX2^+^ cells are distributed in the optic tectum and thalamus.** (**A**) Representative images of a tadpole (left) and whole-mount immunofluorescent staining with an anti-SOX2 antibody (right) at stage 49 *Xenopus*. The red square indicates the whole optic tectum and the thalamus. Scale bar: 100 μm. (**B**) Representative images showing the colabeling with SOX2 and Nkx2.2 in the brain. The dotted white line indicates the outline of the optic tectum (OT). The white line represents the boundary of the SOX2 and Nkx2.2 immunoreactive thalamus (Th). Scale bar: 50 μm. (**C**) Six representative coronal planes of the whole brain with SOX2 immunostaining are shown at stage 49 *Xenopus* (**Ca**–**Cf**). Scale bar: 20 μm. (**D**) One representative sagittal section was immunostained with an anti-SOX2 antibody. The white lines (a–e) depict the positions of coronal sections for (**Ca**–**Ce**). Scale bar: 100 μm. III: third ventricle; IV: fourth ventricle; A: anterior; ABB: alar basal boundary; BH: basal hypothalamus; c-Th: caudal thalamus; D: dorsal; Hb: habenula; M: middle ventricle; Mes: mesencephalon; oc: optic chiasm; P: posterior; p1–3: prosomere1–3; Pa: pallium; PTh: prethalamus; r1–r7: rhomeres1–7; Spa: subpallium; OT: optic tectum; R: rostral; r-Th: rostral thalamus; Th: thalamus; Zli: zona limitans intrathalamica.

**Figure 2 ijms-24-13593-f002:**
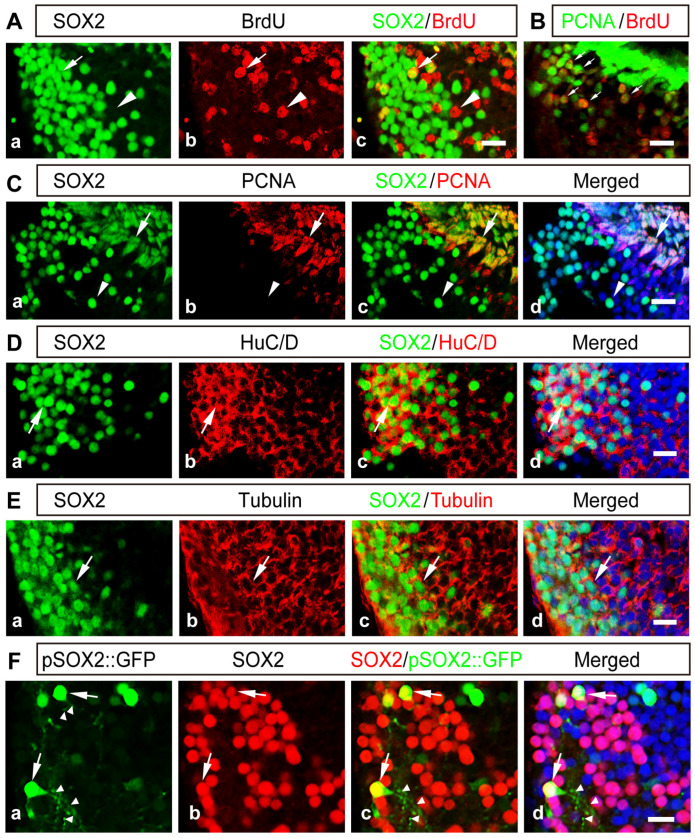
**The majority of thalamic SOX2^+^ cells are HuC/D^+^ or tubulin^+^ neurons.** (**A**) Colabeling of SOX2 and BrdU showing that only a few SOX2^+^ cells (**Aa**) are BrdU^+^ cells (**Ab**) in the thalamus. Arrowheads indicate the SOX2^−^ and BrdU^+^ cells (**Ac**). Arrows indicate the SOX2^+^ and BrdU^+^ cells (**Ac**). Scale bar: 20 μm. (**B**) Coimmunostaining of BrdU and PCNA in the thalamus. Arrows indicate the BrdU^+^ and PCNA^+^ cells. Scale bar: 20 μm. (**C**) Coimmunostaining of SOX2 (**Ca**) and PCNA (**Cb**) in the thalamus. Arrowheads indicate the SOX2^+^ and PCNA^−^ cells. Arrows indicate the SOX2^+^ and PCNA^+^ neurons (**Cc**,**Cd**). Scale bar: 20 μm. (**D**) Coimmunostaining of SOX2 (**Da**) and HuC/D (**Db**) in the thalamus. Arrows indicate the SOX2^+^ and HuC/D^+^ neurons (**Dc**,**Dd**). Scale bar: 20 μm. (**E**) Colabeling of SOX2 (**Ea**) and tubulin (**Eb**) in the thalamus. Arrows indicate the SOX2^+^ and tubulin^+^ neurons (**Ec**,**Ed**). Scale bar: 20 μm. (**F**) Thalamic cells were immunostained with an anti-SOX2 antibody (red, **Fb**) followed by transfection with pSOX2::GFP (green, **Fa**), showing that the SOX2^+^ cells exhibited neuronal morphology with predicted dendrites (arrowheads, **Fc**,**Fd**). Arrows indicate the GFP–expressing and SOX2–immunoreactive cells. Scale bar: 20 μm.

**Figure 3 ijms-24-13593-f003:**
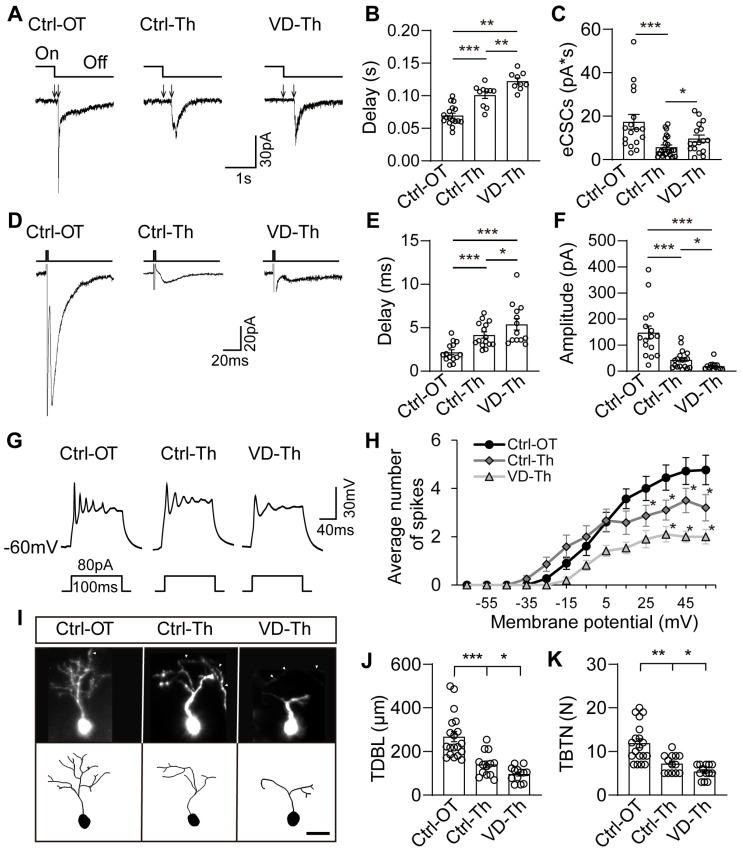
**Visual deprivation alters synaptic transmission and dendritic growth.** (**A**) Representative electrophysiological recordings of visual stimuli–evoked excitatory compound synaptic currents (eCSCs) in Ctrl-OT, Ctrl-Th, and VD-Th neurons in response to full–field light ON and OFF visual stimuli at an intensity of 20 cd/cm^−2^. Arrows indicate the onset and offset of the delay. Scale bar: 30 pA, 1 s. (**B**,**C**) Statistical results show the delay (**B**) and charge transfer (**C**) of eCSCs in Ctrl-OT, Ctrl-Th, and VD-Th neurons. White circles indicate the individual data. N = 17, 28, 17 for Ctrl-OT, Ctrl-Th, and VD-Th groups. (**D**) Representative recordings of optic chiasm stimuli–evoked excitatory postsynaptic currents (EPSCs). Scale bar: 20 pA, 20 ms. (**E**,**F**) Statistical results show the delay (**E**) and amplitude (**F**) of EPSCs. N = 14, 15, 13 for Ctrl-OT, Ctrl-Th, and VD-Th groups. (**G**) Three representative recording traces show current injection–induced spikes in Ctrl-OT, Ctrl-Th, and VD-Th neurons. Scale bar: 30 mV, 40 ms. (**H**) Statistical results show that the number of action potentials was significantly decreased in VD-Th neurons compared to Ctrl-Th neurons. N = 21, 16, 19 for Ctrl-OT, Ctrl-Th, and VD-Th groups. (**I**) Three representative neurons (**upper panel**) and their reconstructed images (**lower panel**) show pSOX2::GFP–expressing neurons in Ctrl-OT, Ctrl-Th, and VD-Th groups. Arrowheads indicate axons. Scale bar: 10 μm. (**J**) Total dendritic branch length (TDBL) was significantly decreased over 48 h in VD-Th neurons compared to Ctrl-Th neurons. (**K**) Total branch tip number (TBTN) was significantly decreased in VD-Th neurons compared to Ctrl-Th neurons. N = 19, 13, 13 for Ctrl-OT, Ctrl-Th, and VD-Th groups. * *p* < 0.05, ** *p* < 0.01, *** *p* < 0.001.

**Figure 4 ijms-24-13593-f004:**
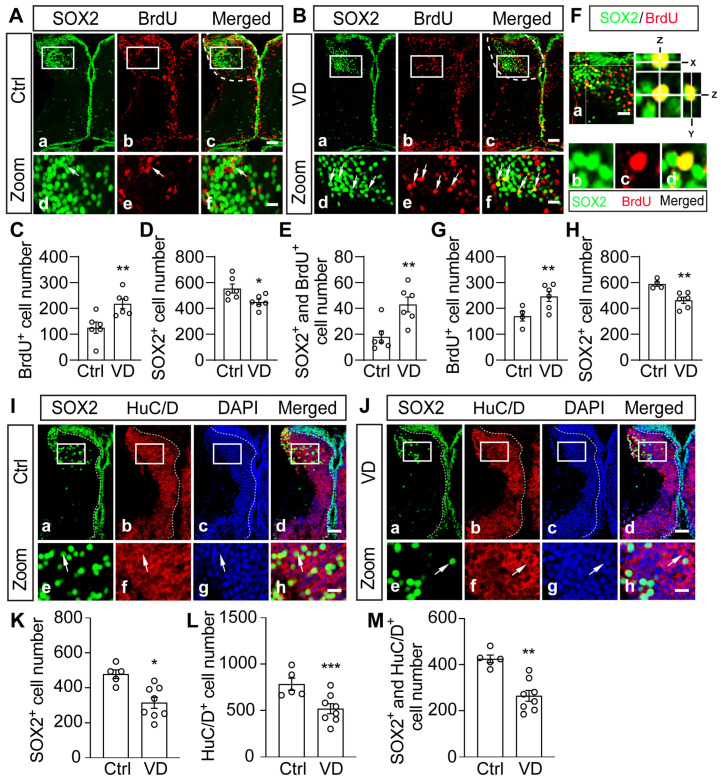
**Visual deprivation changes the balance between proliferation and differentiation in the thalamus.** (**A**,**B**) Representative fluorescent images showing BrdU– (**Ab**) and SOX2–labeled (**Aa**) cells in Ctrl (**Aa**–**Ac**) and VD (**Ba**–**Bc**) thalamus. The white square indicates the BrdU– and SOX2–labeled cells in the zoomed–in thalamus (**Aa**–**Ac**). White dotted lines indicate the boundary of the thalamus (**Ac**). Arrows indicate the SOX2^+^ and BrdU^+^ cells (**Ad**–**Af**,**Bd**–**Bf**). Scale bar: 50 μm. Zoom Scale bar: 20 μm. (**C**–**E**) Summary data show that VD increased BrdU^+^ cells (**C**), decreased SOX2^+^ cells (**D**), and increased SOX2^+^/BrdU^+^ cells (**E**). N = 6, 6 for Ctrl and VD. (**F**) A representative immunofluorescent image showing z–stack for SOX2– and BrdU–labeled cells. Scale bar: 20 μm. (**G**,**H**) Summary data showing that VD increased BrdU^+^ cells (**G**) and decreased SOX2^+^ cells (**H**) in VD–treated tadpoles. N = 4, 6 for Ctrl and VD. (**I**,**J**) Representative immunofluorescent images showing SOX2– and HuC/D–labeled cells in the thalamus of Ctrl (**Ia**–**Id**) and VD (**Ja**–**Jd**) tadpoles. The white square indicates the zoomed–in images. Arrows indicate the SOX2^+^ and HuC/D^+^ cells (**Ie**–**Ih**,**Je**–**Jh**). Scale bar: 50 μm. Zoom scale bar: 20 μm. (**K**–**M**) Summary of data showing that VD decreased HuC/D^+^, SOX2^+^, and HuC/D^+^/SOX2^+^ cells. N = 5, 8 for Ctrl and VD. * *p* < 0.05, ** *p* < 0.01, *** *p* < 0.001.

**Figure 5 ijms-24-13593-f005:**
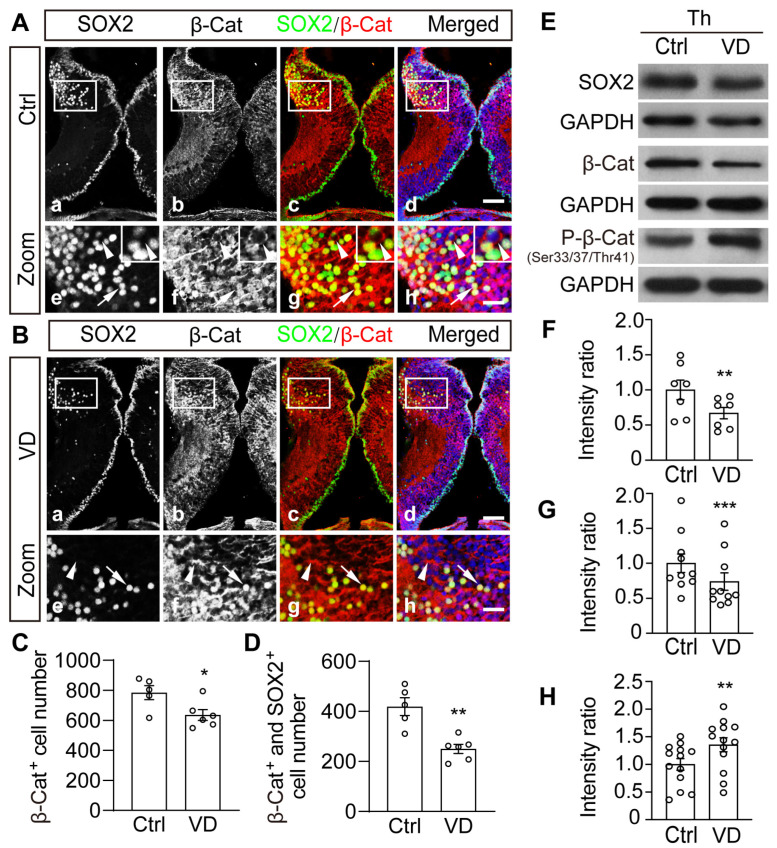
**Visual deprivation reduces nuclearized β-catenin and differentiated neurons in the thalamus.** (**A**,**B**) Representative fluorescent images show β-catenin– and SOX2–labeled cells in the Ctrl (**Aa**–**Ad**) and VD (**Ba**–**Bd**) thalamus. The white square indicates the β-catenin– and SOX2–labeled cells in the zoom of the thalamus. Arrows indicate the expressions of β-catenin– and SOX2 in the nuclei. Arrowheads indicate that β-catenin was expressed in the cytoplasm (**Ae**–**Ah**,**Be**–**Bh**). Scale bar: 50 μm, zoom scale bar: 20 μm. (**C**,**D**) Summary data show that VD decreased β-catenin nuclear localization in labeled cells. N = 5, 6 for β-catenin and SOX2. (**E**) Western blot analysis of homogenates from Ctrl and VD–treated brains using the anti-SOX2, anti-β-catenin, or anti-Phospho-β-catenin (P-β-Cat) antibody. (**F**–**H**) Summary of data showing the relative intensities of SOX2 ((**F**), N = 7), β-catenin ((**G**), N = 10), and P-β-Cat ((**H**), N = 13) to GAPDH. * *p* < 0.05, ** *p* < 0.01, *** *p* < 0.001.

**Figure 6 ijms-24-13593-f006:**
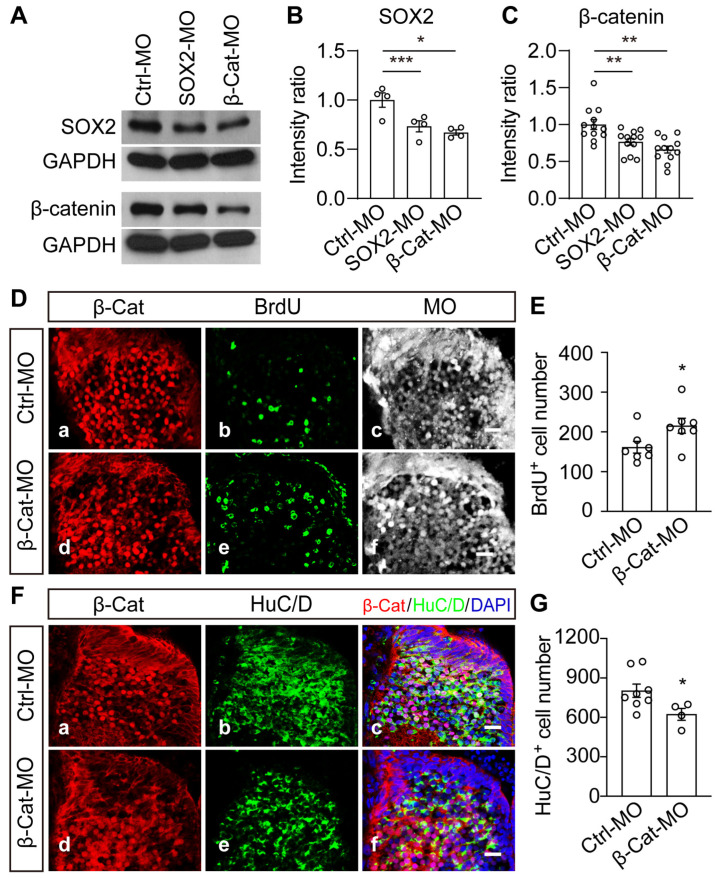
**SOX2-MO and β-Cat-MO knockdown decreased β-catenin and SOX2 expression.** (**A**) Western blot analysis of homogenates from Ctrl-MO–, SOX2-MO– and β-Cat-MO–transfected brains using the anti-SOX2 or anti-β-catenin antibody. (**B**) Quantification results show that SOX2 expression was significantly decreased in SOX2-MO or β-Cat-MO expressing cells compared to Ctrl-MO expressing cells. N = 4. (**C**) Summary of data showing that the relative intensity of the β-catenin group was significantly decreased compared to that of the Ctrl-MO group. N = 12. (**D**) Representative fluorescent images showing the immunostaining of β-catenin and BrdU in the Ctrl-MO– (**Da**–**Dc**) or β-Cat-MO–transfected (**Dd**–**Df**) cells. Scale bar: 20 μm. (**E**) Summary data show that the knockdown of β-catenin increased the number of BrdU^+^ cells. N = 7, 7 for Ctrl-MO and β-Cat-MO. (**F**) Representative images (**Fa**–**Ff**) showing the immunostaining of β-catenin (**Fa**,**Fd**) and HuC/D (**Fb**,**Fe**). Scale bar: 20 μm. (**G**) Summary data show that the knockdown of β-catenin decreased the number of HuC/D^+^ cells. N = 8, 4 for Ctrl-MO and β-Cat-MO. * *p* < 0.05, ** *p* < 0.01, *** *p* < 0.001.

**Figure 7 ijms-24-13593-f007:**
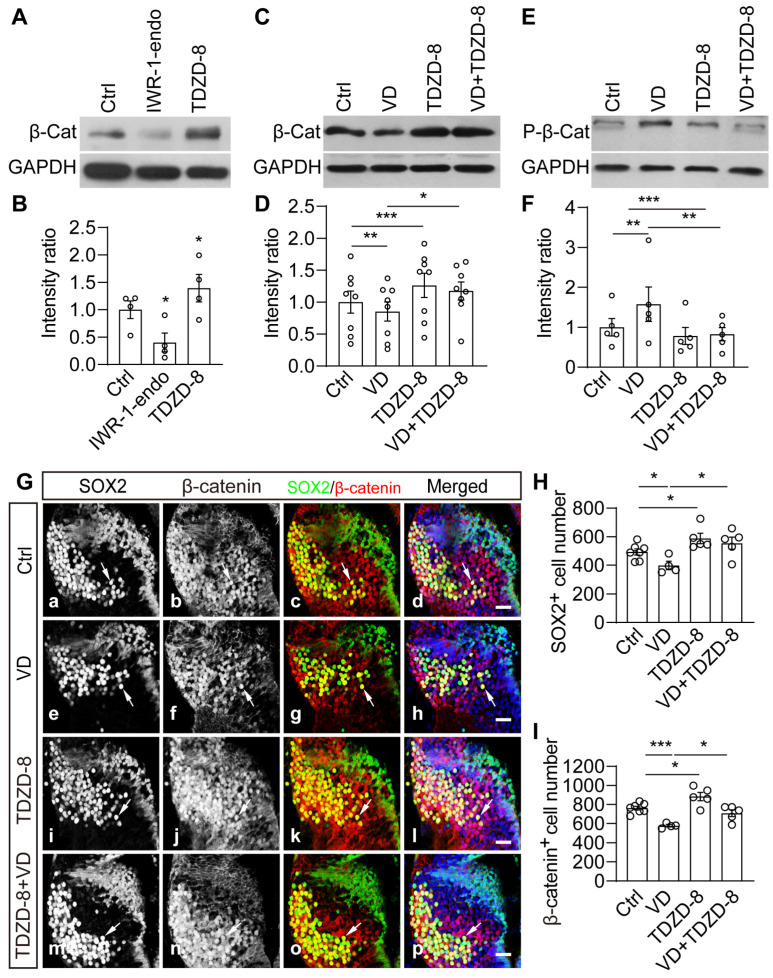
**The visual deprivation-induced decrease in SOX2^+^ and β-catenin^+^ cells was prevented by TDZD-8.** (**A**,**B**) Western blot analysis showing that the relative intensity of β-catenin to GAPDH was significantly decreased by IWR-1-endo but increased by TDZD-8. N = 4. (**C**,**D**) Western blot analysis showing that VD–induced decrease in β-catenin expression was blocked by TDZD-8 treatment. N = 8. (**E**,**F**) VD–induced increase in P-β-Cat was prevented by TDZD-8 treatment. N = 5. (**G**) Representative immunofluorescent images showing SOX2– and β-catenin–labeled cells in the thalamus of Ctrl (**Ga**–**Gd**), VD (**Ge**–**Gh**), TDZD-8 (**Gi**–**Gl**), and VD + TDZD-8 (**Gm**–**Gp**) tadpoles. Arrows indicate the double–labeling cells in the thalamus. Scale bar: 20 μm. (**H**,**I**) Summary of data showing that VD decreased SOX2^+^ (**H**) and β-catenin^+^ (**I**) cells. TDZD-8 prevented the VD–induced decrease in SOX2^+^ and β-catenin^+^ cells in the thalamus. N = 7, 4, 5, 5 for Ctrl, VD, TDZD-8, VD + TDZD-8. * *p* < 0.05, ** *p* < 0.01, *** *p* < 0.001.

**Figure 8 ijms-24-13593-f008:**
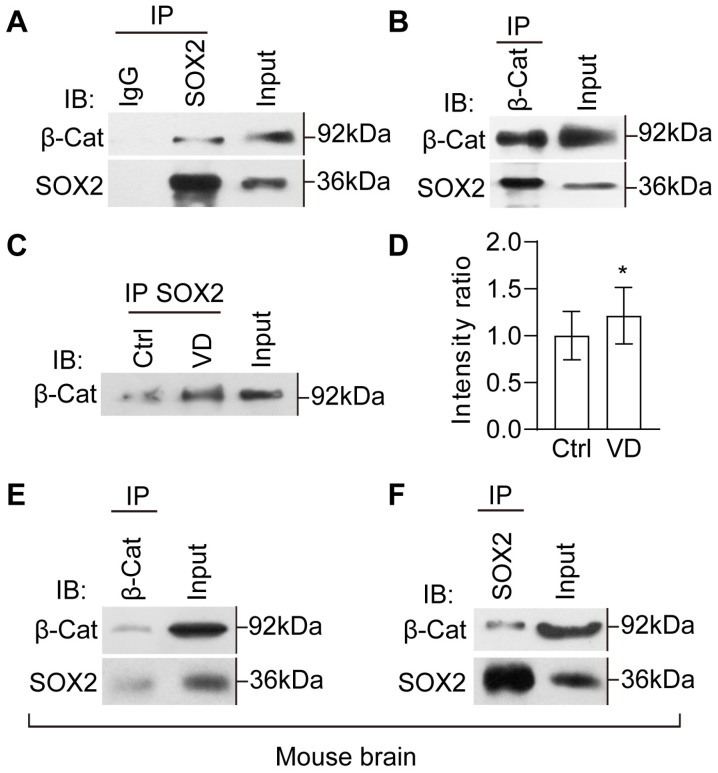
**SOX2 is selectively associated with β-catenin.** (**A**) Coimmunoprecipitation of SOX2 with β-catenin. N = 3 experiments. (**B**) Coimmunoprecipitation of β-catenin with SOX2. N = 4 experiments. (**C**) Coimmunoprecipitation of SOX2 with β-catenin in Ctrl and VD–treated thalamus. (**D**) Summary of data showing that VD increases the interaction between SOX2 with β-catenin. N = 6. * *p* < 0.05. (**E**,**F**) Coimmunoprecipitation in homogenates from the mouse thalamus. Coimmunoprecipitation of β-catenin with SOX2 (**E**). Coimmunoprecipitation of SOX2 with β-catenin (**F**). N = 2 experiments.

**Table 1 ijms-24-13593-t001:** **List of** **antibodies.**

Antigen, Host Species	Immunogen	Source	Catalog No.	RRID	Dilution
β-catenin, mouse	C-terminus of human β-catenin	CST (Danvers, MA, USA)	2677	AB_1030943	1:200 (IF)
					1:50 (IP)
					1:1000 (WB)
β-catenin, rabbit	Residues surrounding Pro714 of human β-catenin protein	CST	8480	AB_11127855	1:200 (IF)
					1:50 (IP)
					1:2000 (WB)
BLBP, mouse	Amino acids 1–132 of human BLBP	Abcam (Cambridge, UK)	ab131137	AB_11157091	1:100 (IF)
BrdU, mouse	BrdU conjugated to KLH	Sigma (St. Louis, MO, USA)	B2531	AB_476793	1:100 (IF)
GAPDH, rabbit	C-terminus of human GAPDH	Millipore (Burlington, MA, USA)	ABS16	AB_11211543	1:10,000 (WB)
HuC/D, mouse	Recombinant human HuC/HuD	Thermo Fisher (Waltham, MA, USA)	A-21271	AB_221448	1:50 (IF)
					1:1000 (WB)
Nkx2.2, mouse	Chicken Nkx2.2	DSHB (Iowa City, IA, USA)	74.5A5	AB_531794	1:50 (IF)
PCNA, rabbit	Recombinant human PCNA	Abcam	ab18197	AB_444313	1:200 (IF)
Phospho-β-catenin	Residues surrounding Ser33, Ser37 and Thr41 of human β-catenin	CST	9561	AB_331729	1:1000 (WB)
SOX2, mouse	Residues surrounding Gly179 of human SOX2 protein	CST	4900	AB_10560516	1:200 (IF)
					1:1000 (WB)
					1:50 (IP)
SOX2, rabbit	Recombinant human SOX2	Abcam	ab97959	AB_2341193	1:200 (IF)
					1:2000 (WB)
					1:50 (IP)
SOX9, rabbit	Recombinant human SOX9	Abcam	ab185230	AB_2715497	1:200 (IF)
Tubulin, mouse	C-terminal of mouse α-tubulin	Beyotime (Shanghai, China)	AT819		1:200 (IF)
Vimentin, rabbit	Recombinant human vimentin	Abcam	ab16700	AB_443435	1:200 (IF)

## Data Availability

Data supporting the findings of this study shall be made available in the article and the Appendix A or from the corresponding authors upon reasonable request.

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
