# Peer review of "β-Catenin and SOX2 Interaction Regulate Visual Experience-Dependent Cell Homeostasis in the Developing *Xenopus* Thalamus"

_ijms, 2023, doi:10.3390/ijms241713593_

Round 1
Reviewer 1 Report
The manuscript submitted to IJMS by Juanmei Gao, et al., describes the vertebrate brain where sensory experience shapes the thalamocortical connections which are critical for visual processing. Authors describe that lacuna lies in how visual experience influences tissue homeostasis and neurogenesis in the developing thalamus. To accomplish this the authors, use an established model system in their lab – Xenopus. The results suggest that β- catenin interacts with SOX2 and is involved in thalamus development. To give a better understanding of this extensive experiments on the mammalian model would help.
1) The experiments using Xenopus could be repeated in the mouse brain especially to show β- catenin interacts with SOX2 to maintain homeostatic neurogenesis.
2) The introduction can be better structured to give a glimpse of the lacuna in the field. Including some of the relevant citations would come in handy in this regard ex: Sara Mercurio, Et al., Cells – MDPI, 2022,
3) Image analysis would provide more insights on SOX2+ cells distributed in the optic tectum in Fig 1 and the same is true for thalamic SOX2+ cells co-localization characterized in Fig 2.
4) Page 19- Materials and methods - It has been observed that in case of the Figure 8, the inputs that are shown are of the mouse brain. But there hasn’t been any information about the mouse brain in the Materials section. It would have been better if you provide brief information about which part of the mouse brain you are considering and its significance in the experiment, since it would be easy for the readers to understand the figure and the concept more, otherwise it is likely to create a state of confusion. It would also be important to mention the animal ethics for the procurement and maintenance of mice to be mentioned in the materials section.
5) Page 17- 330,335,336 - An explanation of figure 10 is provided here but there is no corresponding figure labelled as 10. It would have been better if you provide the figure number and their explanation correctly so that the readers won’t get confused while understanding the figures.
6) Page 2- 66-70 - The sentences in these lines do not clearly represent an overall view of the concept. A little more detailed explanation would help introduce the concept better.
7) Page 2 – 80 - Certain proteins such as Nkx2.2, SOX9 were not elaborated in the prior section. It would have been better if you provide a brief introduction about the proteins before the actual experiments so that it would help the readers to link the mechanisms which you are trying to convey to them.
8) Page 17- Fig -8A - The blots for the input controls have not been put up. This might help in a better understanding of the data.
9) Page 17- Fig- 8C - In the figure it is mentioned that it is the input of the SOX2, but the SOX2 blots have not been included in the figure. It would have been better to include the blot of SOX2 since it is mentioned, and it would help in better interpretation.
Minor edits, nothing major
Author Response
The manuscript submitted to IJMS by Juanmei Gao, et al., describes the vertebrate brain where sensory experience shapes the thalamocortical connections which are critical for visual processing. Authors describe that lacuna lies in how visual experience influences tissue homeostasis and neurogenesis in the developing thalamus. To accomplish this the authors, use an established model system in their lab – Xenopus. The results suggest that β-catenin interacts with SOX2 and is involved in thalamus development. To give a better understanding of this extensive experiments on the mammalian model would help.
Response: We appreciate the reviewer for the helpful comments on our work. Those comments are valuable for improving our paper. We have read the comments carefully and made essential revisions, and we hope our efforts will meet the standard for publication in this journal. For the extensive experiments on mammalian model, please refer to our response to #1 question.
1) The experiments using Xenopus could be repeated in the mouse brain especially to show β-catenin interacts with SOX2 to maintain homeostatic neurogenesis.
Response: We value the insightful suggestions provided by the reviewer. We agree with the reviewer that utilizing a widely accepted mouse model could enhance our comprehension of the conservation of brain development across various species. In the current study, we have established a Xenopus model in studying visual experience-dependent neurogenesis in the developing vertebrate brain. Leveraging the advantages of Xenopus as a model system, we have extensively investigated the homeostatic regulation of thalamic development through in vivo recordings and imaging, an endeavor that proves challenging with the study of a mouse model. To highlight the focus of our experiment, we have added the species of Xenopus to this manuscript title, showing that our current major purpose is to explore the developmental mechanism in the developing Xenopus thalamus. Previous studies have shown that SOX2 acts in thalamic neurons to control the connectivity of retina-thalamus-cortex in the postmitotic thalamic projection neurons (iScience. 2019,15:257-273. doi: 10.1016/j.isci.2019.04.030). The valuable complementary coimmunoprecipitation experiments conducted in the mice demonstrate the conservative interactions between β-catenin and SOX2 (Figure 8E-F) in the thalamus. It is reasonable to expect that the interaction might also play a critical role in regulating the development and function of thalamus in mice. In our upcoming research, we are interested in studying deeper into the mechanism governing homeostatic neurogenesis in other species including mouse thalamus, further elucidating the evolutionary mechanism of thalamic development.
2) The introduction can be better structured to give a glimpse of the lacuna in the field. Including some of the relevant citations would come in handy in this regard ex: Sara Mercurio, Et al., Cells – MDPI, 2022,
Response: We appreciate the reviewer's thoughtful reminder. We have made significant revisions to the introduction as recommended. Additionally, we have incorporated the suggested citation (Cells. 2022;11(10):1604.) into the manuscript. For further details, please refer to the red-marked words in the revised version.
3) Image analysis would provide more insights on SOX2+ cells distributed in the optic tectum in Fig 1 and the same is true for thalamic SOX2+ cells co-localization characterized in Fig 2.
Response: We thank the reviewer’s suggestion. Figures 1 and 2 have been primarily used for qualitative analysis of the distribution of SOX2 in the brain. Our primary objective was to establish SOX2 as a reliable neuronal marker for thalamic neurons, and we have successfully achieved this goal. In order to enrich the information conveyed, we have added more descriptions accompanying both Figure 1 and Figure 2 in the results section.
4) Page 19- Materials and methods - It has been observed that in case of the Figure 8, the inputs that are shown are of the mouse brain. But there hasn’t been any information about the mouse brain in the Materials section. It would have been better if you provide brief information about which part of the mouse brain you are considering and its significance in the experiment, since it would be easy for the readers to understand the figure and the concept more, otherwise it is likely to create a state of confusion. It would also be important to mention the animal ethics for the procurement and maintenance of mice to be mentioned in the materials section.
Response: We appreciate the reviewer’s reminder. Within the revised results section, we clearly mentioned our utilization of the mouse thalamus to investigate the interactions between β-catenin and SOX2. In response to your suggestion, we have incorporated the specific procedural details for manipulating the mice into the materials section. The use of animals was in accordance with the approval granted by the Animal Ethics Committee of Hangzhou Normal University. For experimental purposes, we dissected the thalamus from C57BL/6 mice on postnatal day (P12). The following details have now been included in the revised version of the manuscript for clarity.
C57BL/6 laboratory mice were kept in a controlled environment with a regulated temperature of 22 ± 1 °C and a 12-hour light/dark cycle. The mice's overall health was monitored daily throughout the study. For the experiments, the mice were humanely euthanized using Avertin-induced deep anesthesia. The thalamus tissue was carefully extracted in accordance with the mice brain map. All procedures involving animals adhered to ethical guidelines and were approved by both the Laboratory Animal Center and the Animal Ethics Committee of Hangzhou Normal University, China ( permit number 2022-1063, issued on March 3, 2022).
5) Page 17- 330,335,336 - An explanation of figure 10 is provided here but there is no corresponding figure labelled as 10. It would have been better if you provide the figure number and their explanation correctly so that the readers won’t get confused while understanding the figures.
Response: Thank you for bringing this to our attention. The number was indeed mislabeled. In the revised manuscript, we have rectified the labeling from figure 10 to now be referred to as figure 8. Your reminder is greatly appreciated.
6) Page 2- 66-70 - The sentences in these lines do not clearly represent an overall view of the concept. A little more detailed explanation would help introduce the concept better.
Response: We are grateful for the reviewer’s suggestion. We have rephrased the main findings of our study. The revised sentence has been marked as red. Please refer to the revised manuscript for more details.
7) Page 2 – 80 - Certain proteins such as Nkx2.2, SOX9 were not elaborated in the prior section. It would have been better if you provide a brief introduction about the proteins before the actual experiments so that it would help the readers to link the mechanisms which you are trying to convey to them.
Response: We have expanded the descriptions pertaining to Nkx2.2 and SOX9 in the introduction section. Additionally, we have incorporated relevant commentary in the discussion section.
8) Page 17- Fig -8A - The blots for the input controls have not been put up. This might help in a better understanding of the data.
Response: The co-immunoprecipitation experiments illustrated in Figure 8A included both a positive and a negative control. For the positive control, we used the whole lysate and blotted it with β-catenin, designated as "Input." As part of the negative control, the homogenates underwent co-immunoprecipitation with IgG, followed by blotting with an anti-β-catenin antibody. We have taken your feedback into consideration and have expanded upon these details in the revised manuscript for a more comprehensive understanding.
9) Page 17- Fig- 8C - In the figure it is mentioned that it is the input of the SOX2, but the SOX2 blots have not been included in the figure. It would have been better to include the blot of SOX2 since it is mentioned, and it would help in better interpretation.
Response: We appreciate the reviewer’s comments. To present the results accurately, we have changed the sentence to “We found that the intensity of β-catenin was significantly increased in VD tadpoles compared to control tadpoles”.
Reviewer 2 Report
In the submitted article, the authors investigate the role of visual deprivation (VD) - one sensory experience - for the development of thalamocortical circuit development in Xenopus laevis. The authors find that VD leads to increased neuronal precursors proliferation at the expense of differentiation, accompanied by a reduction of nuclear beta-catenin. Moreover, they find that SOX2 and beta-catenin display physical interaction supporting an interplay between canonical Wnt signaling and the neurogenic functions of SOX2.
The article is interesting and well executed. I suggest below a series of criticisms which, if addressed, would in my opinion improve the study and its conclusions.
1) At lines 106-107, the authors write "most BrdU+ cells are colocalized with PCNA+ cells (Fig. 2B), indicating that BrdU can be used as a differentiating marker in the thalamus". I partially disagree, as I think that this indicates these might be proliferating non-differentiating cells. Or did the authors use the term “differentiating” to indicate that this marker allows distinguishing between cell populations? Could the authors clarify?
2) The experiment employing the transfection of the pSOX2::GFP reporter [48] is not clear. This reporter was constructed using repeats of the SOX2/OCT binding domains, hence expression of GFP relies on the presence of endogenous SOX2 and/or OCT3/4 proteins. It does not report only SOX2 positive cells. Could the authors clarify this part?
3) Figure 3I is an interesting schematic, but it would be important to see the real data underlying these changes in morphology and dendritic growth.
4) The data on nuclear beta-catenin are not fully persuasive. It would be useful to see the different individual western blots that are used for the quantification presented in Figure 5F-H, perhaps as supplementary image. The single WB shown in Fig 5E is good but present loading controls (GAPDH) quite saturated and leave me the doubt that the relatively small difference observed in total and phospho-beta-catenin are meaningful. Perhaps also seeing additional stainings on Xenopus, rather than a single image, would increase my confidence in the finding. This is all the more relevant considering that observing nuclear beta-catenin via antibody staining has been historically difficult.
5) In describing Figure 6A-C the authors write “SOX2 expression was reduced when β-catenin was knocked down (Fig. 6A-C), implying that β-catenin is required for SOX2 expression. Interestingly, SOX2 knockdown also reduced the expression of β-catenin (Fig. 6A-C), suggesting that a potential feedback mechanism controls the SOX2 expression”. While this would be a beautiful explanation, I am not fully convinced this is supported by this single data point, for the reason explained above in my comment 4. A saturated GAPDH decreases the confidence that small differences in band intensities are real. Moreover, I do not really observe a decreased beta-catenin band in SOX2-MO; and if they do, I’d urge them to consider that, in this blot, the intensity of GAPDH in the first lane seems to be slightly higher than that in the central and right lanes.
6) The rescue experiment of Figure 7 is beautiful, and strongly suggests that beta-catenin mediates the effects of VD. I wonder, however, if the authors can find other readouts to show that beta-catenin can rescue the proliferation/differentiation imbalance induced by VD. Have they considered looking at later stages of tadpole? Can sight ability be measured, perhaps in adult animals? A functional experiment, showing the consequences on vision, rather than only a slight difference in cell proliferation, would enormously broaden the reach of the conclusions.
Minor points:
- Line 66: correct "evolutionally"
- Line 71: please write the extended version of BLBP+
- Line 103: I suggest rephrasing into "SOX2+ cells incorporated less BrdU"
- It would be useful if the authors justified (with references) the choice of Nkx2.2 (Figure 1) and of PAX7 (Figure 2).
I made a few suggestions to improve some part, but overall the language is clear.
Author Response
In the submitted article, the authors investigate the role of visual deprivation (VD) - one sensory experience - for the development of thalamocortical circuit development in Xenopus laevis. The authors find that VD leads to increased neuronal precursors proliferation at the expense of differentiation, accompanied by a reduction of nuclear beta-catenin. Moreover, they find that SOX2 and beta-catenin display physical interaction supporting an interplay between canonical Wnt signaling and the neurogenic functions of SOX2. The article is interesting and well executed. I suggest below a series of criticisms which, if addressed, would in my opinion improve the study and its conclusions.
Response: We appreciate the reviewer for the helpful comments on our work. The reviewer has pointed out the critical significance of our work to the research field. Those comments are valuable for improving our paper. We have read the comments carefully and made essential revisions, and we hope our efforts will meet the standard for publication in the journal.
1) At lines 106-107, the authors write "most BrdU+ cells are colocalized with PCNA+ cells (Fig. 2B), indicating that BrdU can be used as a differentiating marker in the thalamus". I partially disagree, as I think that this indicates these might be proliferating non-differentiating cells. Or did the authors use the term “differentiating” to indicate that this marker allows distinguishing between cell populations? Could the authors clarify?
Response: We appreciate the reviewer’s useful comment. We have correct the mistake in the revised manuscript. The word “differentiating” has been changed to “proliferating”.
2) The experiment employing the transfection of the pSOX2::GFP reporter [48] is not clear. This reporter was constructed using repeats of the SOX2/OCT binding domains, hence expression of GFP relies on the presence of endogenous SOX2 and/or OCT3/4 proteins. It does not report only SOX2 positive cells. Could the authors clarify this part?
Response: We agree with the reviewer's suggestion. Jennifer E. Bestman et al., in their publication, initially employed this construct to mark Sox2-expressing radial glia cells serving as neural progenitors in the developing central nervous system of Xenopus tadpoles. Our principal intention in utilizing the construct was to increase the probability of infecting thalamic cells that express SOX2. Furthermore, we performed post-immunostaining using an anti-SOX2 antibody to verify that the cells labeled with pSOX2::GFP indeed correspond to SOX2-positive cells. As SOX2 cells are localized distinctly within the thalamus, this approach facilitates efficient identification of GFP-positive cells and empowers us to carry out thorough imaging analysis.
3) Figure 3I is an interesting schematic, but it would be important to see the real data underlying these changes in morphology and dendritic growth.
Response: We thank the reviewer’s suggestion. The real neurons have been added to the revised figure.
4) The data on nuclear beta-catenin are not fully persuasive. It would be useful to see the different individual western blots that are used for the quantification presented in Figure 5F-H, perhaps as supplementary image. The single WB shown in Fig 5E is good but present loading controls (GAPDH) quite saturated and leave me the doubt that the relatively small difference observed in total and phospho-beta-catenin are meaningful. Perhaps also seeing additional stainings on Xenopus, rather than a single image, would increase my confidence in the finding. This is all the more relevant considering that observing nuclear beta-catenin via antibody staining has been historically difficult.
Response: We greatly appreciate the insightful input provided by the reviewer. In an effort to optimize the efficiency of Western blotting, we implemented a strategy of placing the PVDF membranes with multiple samples within the same developing box and utilizing ECL chemiluminescence for visualization. Given the varying sensitivities of the antibodies used, the resulting bands exhibited diverse degrees of exposure. It's important to note that we conducted multiple exposures by adjusting the membrane exposure times, which led to the generation of blotting bands with different intensities. In response to this valuable feedback, we have integrated the revised figure with updated loading controls. Additionally, as the reviewer's recommendation, we have incorporated individual Western blots into the supplementary figure (Fig. S7) to provide a more comprehensive representation of the experimental results.
5) In describing Figure 6A-C the authors write “SOX2 expression was reduced when β-catenin was knocked down (Fig. 6A-C), implying that β-catenin is required for SOX2 expression. Interestingly, SOX2 knockdown also reduced the expression of β-catenin (Fig. 6A-C), suggesting that a potential feedback mechanism controls the SOX2 expression”. While this would be a beautiful explanation, I am not fully convinced this is supported by this single data point, for the reason explained above in my comment 4. A saturated GAPDH decreases the confidence that small differences in band intensities are real. Moreover, I do not really observe a decreased beta-catenin band in SOX2-MO; and if they do, I’d urge them to consider that, in this blot, the intensity of GAPDH in the first lane seems to be slightly higher than that in the central and right lanes.
Response: We appreciate the reviewer’s suggestion. In order to conduct our comparisons, we utilized the relative intensity of SOX2 in relation to the loading control, GAPDH, rather than relying solely on the absolute values of SOX2 or GAPDH intensities. This approach was adopted to account for potential variability among the loading samples across different experiments. To bolster the strength of our findings, we have included the individual Western blots in the supplementary figure (Fig. S8), as this addition aligns with the reviewer's recommendation.
6) The rescue experiment of Figure 7 is beautiful, and strongly suggests that beta-catenin mediates the effects of VD. I wonder, however, if the authors can find other readouts to show that beta-catenin can rescue the proliferation/differentiation imbalance induced by VD. Have they considered looking at later stages of tadpole? Can sight ability be measured, perhaps in adult animals? A functional experiment, showing the consequences on vision, rather than only a slight difference in cell proliferation, would enormously broaden the reach of the conclusions.
Response: We greatly appreciate the suggestion provided by the reviewer. In this manuscript, Xenopus was used as a vertebrate model to investigate the early stages of brain development. The advantages of utilizing early developing tadpoles are notable, as they offer dynamic insights into nerve cell morphology, electrophysiological recordings, and the development of the nervous system driven by visual experiences. Typically, we focus on the early stages of tadpole development (prior to stage 49) to conduct comprehensive investigations aimed at unraveling the establishment and functional transformations within the thalamus. However, it's important to note that Xenopus tadpoles require a span of 1 to 2 months to initiate metamorphosis, and approximately a year to reach adulthood. Acknowledging the intriguing notion, we concur that exploring the potential impacts of VD on thalamic development and function presents an intriguing avenue. In future studies, we remain open to delving into the phenomena and mechanisms underlying this particular stage of development.
Minor points:
- Line 66: correct "evolutionally"
Response: We have corrected to “evolutionarily”.
- Line 71: please write the extended version of BLBP+
Response: The full name of BLBP was added to the introduction in the first place that BLBP was shown.
- Line 103: I suggest rephrasing into "SOX2+ cells incorporated less BrdU"
Response: We have rephrased the sentence according to the reviewer’s suggestion.
- It would be useful if the authors justified (with references) the choice of Nkx2.2 (Figure 1) and of PAX7 (Figure 2).
Response: We have added the relative references to the introduction. In the result 2.1, Nkx2.2 was shown to label subpopulations of caudal thalamus.
Round 2
Reviewer 1 Report
Thanks for addressing the previous comments. No further comments